# Recent Trends in Hydroxyapatite Supplementation for Osteoregenerative Purposes

**DOI:** 10.3390/ma16031303

**Published:** 2023-02-03

**Authors:** Ana Zastulka, Simona Clichici, Maria Tomoaia-Cotisel, Aurora Mocanu, Cecilia Roman, Cristian-Doru Olteanu, Bogdan Culic, Teodora Mocan

**Affiliations:** 1Physiology Department, Iuliu Hatieganu University of Medicine and Pharmacy, 1 Clinicilor Street, 400006 Cluj-Napoca, Romania; 2Department of Chemical Engineering, Faculty of Chemistry and Chemical Engineering, Research Center in Physical Chemistry, Babes-Bolyai University of Cluj-Napoca, 11 Arany Janos Str., 400028 Cluj-Napoca, Romania; 3Academy of Romanian Scientists, 3 Ilfov Str., 050044 Bucharest, Romania; 4INCDO-INOE 2000, Research Institute for Analytical Instrumentation, 400296 Cluj-Napoca, Romania; 5Orthodontic Department, Iuliu Hatieganu University of Medicine and Pharmacy, 31 Avram Iancu Street, 400083 Cluj-Napoca, Romania; 6Department of Prosthetic Dentistry and Dental Materials, Iuliu Hatieganu University of Medicine and Pharmacy, 32 Clinicilor Street, 400012 Cluj-Napoca, Romania; 7Nanomedicine Department, Regional Institute of Gastroenterology and Hepatology Cluj-Napoca, 5 Constanta Street, 400158 Cluj-Napoca, Romania

**Keywords:** hydroxyapatite, bone cells, signaling, synthesis methods, nanoparticles, curcumin

## Abstract

Bone regeneration has gained attention in the biomedical field, which has led to the development of materials and synthesis methods meant to improve osseointegration and cellular bone activity. The properties of hydroxyapatite, a type of calcium phosphate, have been researched to determine its advantages for bone tissue engineering, particularly its biocompatibility and ability to interact with bone cells. Recently, the advantages of utilizing nanomolecules of hydroxyapatite, combined with various substances, in order to enhance and combine their characteristics, have been reported in the literature. This review will outline the cellular and molecular roles of hydroxypatite, its interactions with bone cells, and its nano-combinations with various ions and natural products and their effects on bone growth, development, and bone repair.

## 1. Introduction

Bone represents a mineralized tissue that possesses a primary role in the support of the human body. Bone defects can be attributed to processes such as aging, inflammation, infections, trauma or tumors, and lead to a significant impact on overall health [1]. Critical bone defects, defined as being more than 2 cm in length or surpassing 50 percent of the circumference of the defect, necessitate the addition of biomaterials for their repair, as they are not otherwise capable of full regeneration [2]. Self-polymerizing biomaterials, known as bone cements, are utilized for bone defect treatment in orthopedics, traumatology, or maxillofacial surgery, after trauma or bone tumor resection [3,4]. Recent trends have focused on the research and development of biomaterials for bone repair that have proven to have high scientific and clinical value. Synthetic calcium phosphates, particularly hydroxyapatite, facilitate bone cell attachment, proliferation and differentiation, and are additionally characterized by biocompatibility, osteoconductivity, and osteoinductivity. Hydroxyapatite is a ceramic bone cement that bears great resemblance to the bone crystalline phase [5]. This review presents an up-to-date literature overview of the various uses of hydroxyapatite as a biocompatible material for bone regeneration, the connection between the roles of bone cells and hydroxyapatite, the role of signaling pathways on bone development, several methods for the synthesis of hydroxyapatite and the various combinations between hydroxyapatite and different substances and their advantages in bone growth and development. The review also aims to provide insight into future bone regeneration mechanisms and pathways. 

## 2. Materials and Methods

In this review, we present an outline of the most notable current trends in bone regeneration, their applications and relevance. We conducted a Medline/PubMed database search for eligible articles by utilizing specific keywords, such as “hydroxyapatite”, “bone cells”, “signaling”, “synthesis methods”, “nanoparticles” and “curcumin”, either together or in different combinations. We analyzed the articles according to their relevance for our desired study and proceeded to add the ones that fulfilled our criteria: the interactions between bone tissue and hydroxyapatite and its varied nano-combinations. Only English-language articles were included in this study; the majority of articles were published between 2002 and 2022, with a few exceptions we were not able to exclude due to their significance. 

## 3. Hydroxyapatite 

### 3.1. Components, Structure and Properties of Hydroxyapatite

Calcium phosphates are types of minerals composed of calcium cations and phosphate anions; namely, orthophosphates, metaphosphates or pyrophosphates. Of particular interest for this review is a naturally occurring form of calcium phosphate, hydroxyapatite, which is the main mineral component of teeth and bones. Hydroxyapatite (HAP) also represents the type of calcium phosphate most often utilized in bone grafting due to its ability to aid in bone regeneration. 

Hydroxyapatite (HAP) is an inorganic substance containing an apatite lattice structure of $A_{10}B_{6}C_{2}$, with the A, B and C structures typically represented by Ca, ${\mathrm{PO}}_{4}$ and OH. Pure HAP is composed of 39.68% by-weight calcium and 18% by-weight phosphorus, leading to a Ca/P molar ratio of 1.67. Commercial HAP products, containing different Ca/P ratios, either bigger or smaller than 1.67, indicate the shift between the tricalcium phosphate (TCP) phase and the calcium oxide (CaO) phase, characterized by a Ca/P ratio larger than 1.67. HAP with a Ca/P ratio bigger than 1.67 comprises more CaO than TCP and vice versa [6,7]. Structurally, HAP is similar to the bone mineral phase, presenting high biocompatibility and bioactive properties [8,9]. It has been proven that the biocompatibility and bioactivity of HAP, represented by osteoconductive and osteoinductive processes, support osseointegration. The osteoconductivity of HAP provides guidance for the synthesis of new bone on its surface [10]. The osteoconductive property allows osteoblasts to attach, proliferate, grow and express the phenotype by direct contact, thereby creating a strong tissue–implant interface. Osteoconduction is reliant upon the specific geometry and size of the HAP pores [10,11]. Alternatively, the osteoinduction of HAP facilitates tissular growth that results in bony neoformation in non-bone-forming sites as well. These properties have led to the extensive use of HAP in scientific fields; namely, chemistry and biology [9], and in medical areas concerning hard tissues, such as traumatology, maxillofacial surgery or dentistry [11,12], and in the science of orbital implants [13].

### 3.2. Bone Cells and Their Interactions with Hydroxyapatite

Bone resorption and apposition are physiological processes determined by the balance between osteoblasts, osteocytes and osteoclasts. The bone possesses a regenerative capacity; thus, the adult skeleton is replaced entirely every ten years. Osteocytes, considered to be mature osteoblasts, originate from mesenchymal stem cells [14]. Osteoblasts have the primary role in the deposition of bone osteoid matrix. As the osteoblast matures, it is transformed into an osteocyte, becoming incorporated within the bone matrix [15]. The activity of osteoblasts and osteoclasts is strongly regulated by signals that originate from osteocytes [16]. Table 1 poses an overview of bone cells types and their roles.

#### 3.2.1. Osteoblasts

Osteoblasts are bone-forming cells that are created from mesenchymal precursors through a sequence of transcriptional factors and afterwards are transformed into osteocytes [17,18,19,20]. Osteoblasts synthesize extracellular proteins, namely, osteocalcin, alkaline phosphatase and type I collagen, the latter of which amounts to more than 90% of bone matrix proteins. The extracellular matrix is firstly secreted in the form of unmineralized osteoid and subsequently undergoes the process of mineralization, through the increase of calcium phosphate, structured as hydroxyapatite [21,22]. Osteoblasts create the collagen bone matrix and play a crucial part in the regulation of its mineralization. The osteoblast lineage is composed of matrix-producing osteoblasts, their pluripotent and lineage-committed precursors, bone lining cells and matrix-embedded osteocytes. Every cellular stage has well-defined markers and individual roles, morphologies and locations relative to the osseous surface [23]. Endogenous HAP is synthesized by osteoblasts as matrix vesicles that initiate the formation of bone in the skeleton. There are several reports about the effects that free nano-HAPs have on cell proliferation, apoptosis and osteoblast differentiation. An in vitro study determined that 20–40 nm apatite particles are a key factor in biomineral formation [24], while another study concluded that nano-HAPs of various sizes and crystallinities possess different abilities for the promotion of osteoblast proliferation and differentiation and play a role in the inhibition of cell apoptosis [25]. Autophagy is a lysosome-based degradative pathway essential to maintaining cellular homeostasis. The role of autophagy in bone cells has been described by several studies. One study highlighted the contribution of autophagy in osteoblast differentiation [26], while another concluded that suppression of autophagy leads to a reduction in osseous volume and contributes to aging in 6-month-old mice [27]. Another study revealed that autophagy in osteoblasts has implications in mineralization and bone homeostasis [28]. Recent developments in nano-biology have demonstrated that different types of nanoparticles are able to induce autophagy, including quantum dots [29], dendrimers [30], fullerene [31], neodymium oxide [32], and gold nanoparticles [33]. A different study highlighted the effect of nano-HAPs on osteoblast differentiation correlated with particle agglomeration and nano-HAP concentration, providing new insight into the biological effects of nano-HAPs on osteoblasts [33].

#### 3.2.2. Osteocytes

Osteocytes comprise more than 95% of bone cells, amounting to a total of approximately 42 billion in the adult human skeleton [34,35,36]. Osteocytes are formed during the process when osteoblasts become embedded within the osteoid matrix. Osteocytes do not undergo cellular division and are capable of living in the bone matrix for decades, their life span being closely linked to the remodeling rate of the bone [37]. Osteocytes have been shown to possess a significant role in bone remodeling, in that they influence the bone-forming osteoblasts and the bone-resorbing osteoclasts. Osteocytes represent a vast cellular network, as they communicate with approximately 50 other osteocytes situated in close proximity to one another. Therefore, any stimulus that reaches the osteocytes triggers them to interact in a direct manner with the bone matrix, impacting the overall integrity of the bone positively or negatively [38].

#### 3.2.3. Osteoclasts

Osteoclasts are large, multinucleate cells that are formed in the migration process of circulating mononuclear hematopoietic progenitor cells to the surface of the bone. Subsequently, the progenitor cells become fused to one another and attach to the bone surface in order to create osteoclasts in an active form. The fusion process is generated by dendritic cell-specific transmembrane proteins (DC-STAMP) [39] and osteoclast-specific transmembrane proteins (OC-STAMP) [40]. Osteoclasts are able to create additional osteoclasts through the fission of cells, characterized by the detachment of the released mononuclear or multinucleated osteoclast, and afterwards by the connection to another osteoclast or osteoclast precursor, in order to continue the resorptive process [41]. Once the creation and attachment to the bone surface are completed, osteoclasts become polarized cells, possessing specific functional domains that are responsible for the attachment, resorption and release of resorption products [37]. Osteoclasts have the primary role in the process of bone repair and remodeling through the production of proteolytic enzymes, which leads to the resorption of bone matrix [42]. The resorptive process utilizes the characteristic ruffled border feature of the osteoclast, thereby increasing the surface area available for ion exchange [37]. 

**Table 1 materials-16-01303-t001:** Bone cells. Types and roles.

Cell Type	Role	Reference
Osteoblasts	Bone formation Synthesis of extracellular proteins Synthesis of endogenous HAP	[17,18,19,20,23] [21,22] [24,25]
Osteocytes	Bone remodeling Influence on bone integrity	[37] [38]
Osteoclasts	Bone repair and remodeling Bone matrix resorption	[37] [42]

### 3.3. Molecular Role of Hydroxyapatite and Signaling Pathways 

Osteoblasts and osteocytes play a key role in the signaling pathway of bone through the secretion of two proteins: receptor activator of nuclear factor kappa-β ligand (RANKL) and osteoprotegerin (OPG). RANKL (gene name: TNFSF11) is a significant member of the TNF superfamily, linked to a variety of cells; namely, osteoblasts, osteocytes, hypertrophic chondrocytes, T cells, mammary gland epithelial cells and stromal cells [43], a significant number of whom are located inside the local osseous environment [23,34,42]. The binding reaction between RANKL and RANK (the receptor for RANKL) is restricted by osteoprotegerin (gene name: TNFRSF11B), which is a soluble receptor with a decoy role, inhibiting the physiological osteoclast formation. Early studies in gene knockout mice revealed that RANKL is crucial for osteoclastogenesis, while osteoprogerin (OPG) is capable of inhibiting it [24,44,45]. Communication between different types of bone cells can happen in two ways: from osteoclasts to the osteoblast lineage and vice versa, and from the osteoblast lineage to the osteoclasts, facilitating their formation and initiating bone resorption. The osteoblast cellular lineage plays a main part in the regulation of osteoclast differentiation, as it produces RANKL and OPG, ensuring an integrated interaction between local, systemic or mechanical stimuli and the bone tissue. In addition, osteoblast lineage cells behave as “reversal” cells during the process of bone remodeling [23]. Osteocytes are currently understood to have a crucial role in the regulation of bone strength due to the signals transmitted to cells on the bone surface [23,25] and due to the modification of their local environment [37,46,47].

The osteoblast lineage also utilizes various RANKL-independent mechanisms to calibrate the amount of bone resorption; namely, signals that have the role of initiating the remodeling phase [26] of stimulating the proliferation of osteoblast precursors [27], or factors that have the role of regulating the activity of mature osteoclasts [28]. Osteoclastogenesis is a complex process which involves various regulators. Studies determined that the fusion of myeloid hematopoietic precursors, in order to form multinucleated osteoclasts, is influenced by two specific cytokines: the macrophage colony stimulating factor (M-CSF), which interacts with the receptor c-fms, and RANKL [29,30]. M-CSF and RANKL play a crucial role in the differentiation of osteoclasts, RANKL having the additional role of promoting the osteoclast maturation. The result of excessive cytokine synthesis is an increase in osteoclast differentiation and abnormal levels of bone resorption [31]. It has been recently proven that in vivo physiological osteoclastogenesis is capable of being induced by membrane-bound RANKL and not by other, soluble forms of RANKL [32], as the interaction between RANK and RANKL is contact-dependent [18,33]. Additionally, several studies have reported that RANKL deletion in mature osteoblasts and osteocytes resulted in diminished osteoclastogenesis and increased bone mass, suggesting the role that the production of RANKL by young osteoblasts and osteocytes might play in the formation of osteoclasts [48,49]. 

## 4. Hydroxyapatite Nanoformulations

### 4.1. Nano-Hydroxyapatite Synthesis Methods

Nanotechnology is utilized as an umbrella term that describes a wide array of techniques for the fabrication of materials and devices at the nano-level. The societal impact of nanotechnology has resulted in the development of various synthesis techniques, with the purpose of producing nano-sized materials, particularly HAP nanoparticles (nHAPs) [9]. Further research into these techniques was subsequently conducted in order to create nHAPs with additional size reduction, increased surface areas and controlled morphology, making feasible the regulation and manipulation of the physical, chemical and surface properties of synthetic nHAPs [50,51,52].

There have been two principal methods for the synthesis of nanoparticles reported in the literature: top-down and bottom-up, respectively. The top-down approach consists of the slicing or successive cutting of a bulk material in order to obtain a nano-sized particle. Conversely, the bottom-up approach is characterized by the build-up of a material from the bottom, atom by atom, molecule by molecule or cluster by cluster. Most published reports have focused on the bottom-up approach in synthesizing nHAPs [9,51].

Several reports have presented the diverse preparation methods of HAP nanostructures, such as wet chemical precipitation, sol–gel process, gel method, spray pyrolysis, hydrothermal synthesis, emulsion or micro-emulsion routes and the microwave method (Table 2) [53,54,55,56,57,58,59,60]. The sol–gel method is beneficial for the synthesis of HAP, as this method features many advantages, chief among them the high product purity and low synthesis temperature [54,61,62,63]. Nevertheless, the microwave technique is advantageous due to its property of the rapid rates of homogenous internal and volumetric heating [64,65,66,67,68,69] and due to its superior ability to achieve high-crystalline nanostructure materials, compared with the conventional heating technique [70]. 

#### 4.1.1. Wet Chemical Precipitation Synthesis

Of the multitude of methods that have been explored in the formation of nHAPs, chemical precipitation synthesis has been distinguished due to its versatility, and therefore has been widely reported in the literature [71,72,73,74,75,76,77,78,79]. Various articles have reported the influence that various synthesis parameters, namely, temperature [72,74,75,76], time [71,72], calcium ion concentration [75], surfactant [77], calcination [71,72] and the utilization of diverse reagents [78], play on the morphological features of nHAPs. It has been observed that nHAP particles formed by chemical precipitation tended to form agglomerates, which could represent, according to Rahaman’s classification [80], clusters of ultra-fine primary particles [70]. Wet chemical precipitation synthesis presents an effective method of controlling the size of the particles. Controlling the particle size of nHAPs is thus of critical importance, particularly if the nanoparticles are designed for blood circulation. Studies have established a linear correlation between temperature and the increase in size of the nHAP particles, which confirm that the particle dimensions can be controlled, making the wet chemical precipitation technique very reliable and feasible [9,79].

#### 4.1.2. Hydrothermal Synthesis

This method is based on the chemical reaction of substances in a sealed solution, heated above ambient temperature and pressure [81]. Hydrothermal synthesis is a simple and effective technique that leads to the formation of fine-grained, pure, well-dispersed, single crystals of nHAP [82]. Previous studies have determined that the nanoparticles obtained by employing the wet chemical precipitation synthesis method, while responsive to the alteration of the synthesis temperature, possessed irregular shapes and poor surface morphology [79]. Conversely, the hydrothermal synthesis technique would be a more advantageous option, as it creates HAP nanoparticles of well-defined sizes and morphology [9,79]. The hydrothermal synthesis process is characterized by rapid fabrication, with relative technical simplicity and increased crystallinity levels [82]. In the hydrothermal method, the pH value and ion concentrations (e.g., ${\mathrm{Ca}}^{2+}$, ${\mathrm{PO}}_{4}^{3-}$) are crucial factors that impact the morphology and the crystal dimension of formed coatings [83]. Therefore, this method shows noteworthy advantages in the formation of micro/nano-structured HAP scaffolds, as it is able to positively influence surface morphology independent of scaffold shape [84,85]. 

#### 4.1.3. Micro-Emulsion Synthesis

Micro-emulsion has been proven to be a technique capable of creating particle sizes in the range of nanometers, with a minimum amount of agglomeration [86,87,88], as opposed to the synthesis techniques presented above, that produce nHAP particles with an agglomeration difficult to control, which is a main disadvantage of the aforementioned methods [86,87]. Particles of nHAP ranging in size from 30 nm to 50 nm, with needle-shaped or spherical morphology, can be created by micro-emulsion synthesis [86]. A micro-emulsion is defined as a transparent, thermodynamically stable solution, comprised of two immiscible liquids, namely water and oil, that can be stabilized by an amphiphilic surface-active agent or surfactant [9].

#### 4.1.4. Sol–Gel Synthesis 

The sol–gel coating technique exhibits numerous advantages compared to other techniques listed, as it possesses the ability to perform at low processing temperature [88], implying cost-effectiveness [89], as well as the potential to produce coatings that benefit from high purity and homogeneity [90]. Additionally, this method is able to create uniform, intimate mixtures of various colloidal oxides on a molecular level and the resulting gel can be easily shaped. Moreover, sol–gel synthesis can facilitate molecular control over the chemical composition, assuring the application of miniscule quantities of different components to the sol and their uniform dispersation [91,92]. Furthermore, the developed sol–gel synthesis procedures are environmentally benign, simple and characterized by high reproducibility. It was demonstrated that the sol–gel synthesis technique is a key mechanism for the formation of calcium hydroxyapatite, tricalcium phosphate, other types of phosphates and various calcium-phosphate-based composites. The preparation of these biomaterials by sol–gel methods exhibited high phase purity of the final products, as well as minor changes in the molecule morphology. The sol–gel synthesis method generally permitted the research and development of biomaterials that possess superior characteristics in terms of biomedical applications. The sol–gel technique method has been determined to be an advantageous way to obtain the homogenous distribution of the nanostructural biocomposites components [93]. 

**Table 2 materials-16-01303-t002:** Hydroxyapatite synthesis methods: advantages and disadvantages.

HA Synthesis Method	Advantages	Disadvantages	Reference
Wet chemical precipitation synthesis	Able to control nHAP particle size Versatile, reliable, feasible	Irregular shape Unsatisfactory surface morphology	[70,71,72,73,74,75,76,77,78,79,80] [9,79]
Hydrothermal synthesis	Rapid fabrication Technical simplicity Increased crystallinity Influences surface morphology independent of scaffold shape	Difficult to control agglomeration	[9,79] [82] [84,85]
Microemulsion synthesis	Creates nanometer-sized particles with minimal agglomeration	Unable to perform at low temperature	[86,87,88]
Sol-gel synthesis	Performs at low temperatures Cost-effective High-purity, homogenous coatings Uniform dispersation		[88,89,90,91,92,93]
Microwave synthesis	Rapid, homogenous internal and volumetric heating High crystallinity		[64,65,66,67,68,69,70]

### 4.2. Nano-Hydroxyapatite Mixed with Ions

HAP is able to improve tissue engineering constructs with diverse functionalities. HAP possesses in the structure of its crystalline lattice a configuration that enables the incorporation of various ions, namely $K^{+}$, ${\mathrm{Ag}}^{+}$, ${\mathrm{Na}}^{+}$, ${\mathrm{Mn}}^{2+}$, ${\mathrm{Ni}}^{2+}$, ${\mathrm{Cu}}^{2+}$, ${\mathrm{Co}}^{2+}$, ${\mathrm{Cr}}^{3+}$, ${\mathrm{Sr}}^{2+}$, ${\mathrm{Ba}}^{2+}$, ${\mathrm{Pb}}^{2+}$, ${\mathrm{Cd}}^{2+}$, $Y^{3+},{\;\mathrm{La}}^{3+}$, ${\mathrm{Fe}}^{2-}$, ${\mathrm{Zn}}^{2+}$, ${\mathrm{Mg}}^{2+}$, ${\mathrm{Al}}^{3+}$, ${\mathrm{Si}}^{4+}$, ${\mathrm{CO}}_{3}^{2-},{\;\mathrm{and}\;F}^{-}$, ${\mathrm{Cl}}^{-}$, ${\mathrm{Br}}^{-}$, $O^{2-}$ and ${\mathrm{OH}}^{-}$, that can substitute ${\mathrm{Ca}}^{2+}$, ${\mathrm{PO}}_{4}^{3-}$ and ${\mathrm{OH}}^{-}$ ions, respectively [94,95,96,97,98,99,100,101], which lead to significant differences in biological and mechanical properties (Figure 1). HAP crystallizes in the form of a structure characterized by a hexagonal shape, featuring the lattice parameters a = b = 0.9418 nm, c = 0.6884 nm, with unit cell volume V = 0.5288 nm^3^ [70,102,103,104]. HAP allows the addition of a large variety of ions, other chemical elements, substances and products to the structure of HAP and, subsequently, the establishment of innovative properties [70].

#### 4.2.1. Magnesium-Nano-Hydroxyapatite

The magnesium (Mg) cation is an important component of the early cartilage and bone structure, its concentration diminishing in the later stages of osseous maturity. Lower than physiological magnesium concentrations have unfavorable effects on every phase of skeletal metabolism, resulting in arrested bone development and maturation, reduced activity of the osteoblast and osteoclast cellular lines, osteopenia and bone fragility [105]. Synthetic forms of Mg-hydroxyapatite are efficiently utilized as artificial bone substitutes, featuring properties such as increased reabsorption potential, and are able to be used as a source of Mg as well. Wet chemical synthesis is an advantageous method for the preparation of apatite powders with low levels of crystallinity and high reactivity, as it is capable of calibrating the chemical precursors and process parameters; namely, temperature, concentrations or maturation time [106]. The synthesis of Mg-substituted hydroxyapatite [107,108] can either make use of the process of immersion of HAP in a solution of Mg nitrate by the exchange between ions [107,109,110,111] or of the mixture between magnesium oxide and HAP powders [112,113]. Mg-HAP showed superior mineralization results when used as a coating on Ti implants compared with pure HAP [114], and was also noted to exhibit a higher degree of osseointegration than native HAP, possibly due to an increase in the rate of resorption [70]. One study presented different synthetic HAPs substituted with various concentrations of Mg (6–14 mol%), created with the wet chemical precipitation method. The study concluded that the 5.7 mol% Mg-doped HAP improved the morphology, granulation, composition, solubility and crystallinity of HAP and revealed no cytotoxicity, carcinogenicity or genotoxicity, subsequently increasing the level of biocompatibility of HAP. Mg-HAP demonstrated higher levels of osteoconductivity than the stoichiometric form of simple HAP and notable in vivo results as a capable bone substitute [113].

#### 4.2.2. Zinc-Nano-Hydroxyapatite

Zinc is an essential micro-element present in the human body which possesses a stimulatory effect on bone metabolism [114]. Zinc (Zn) has the capacity to enhance the processes of bone formation and bone mineralization, which, in turn, leads to the stimulation of collagen production and alkaline phosphatase (ALP) activity. The cellular activities of osteoblasts and osteoclasts have been proven to be positively impacted by the interaction with zinc-related proteins (Figure 2) and, therefore, to result in the decrease of early stage bone resorption and deterioration [115]. Alkaline phosphatase utilizes zinc as a co-factor as the process of bone mineralization gradually increases [116,117,118]. Zn complements the bioactivity of HAP [119], as zinc-enforced tricalcium phosphate is administered in order to promote osteogenesis in osteoporotic bone [120,121], in comparison with simple HAPs [122]. The literature indicates that ZnHAP-NPs are useful for the process of osteoregeneration. The nanoparticles of Zn and HAP are superior because they combine the positive features of HAP particles, which closely resemble the HAP crystals present in the structure of natural bone, and those of zinc particles, which enhance HAP’s characteristics, particularly its anti-inflammatory capacity [123]. Zn deficiency results in skeletal modifications such as arrest of skeletal growth, prolonged bone recovery, diminished bone mass during the premenopausal phase and osteoporosis occurring in the postmenopausal period [114,124,125].

#### 4.2.3. Selenium-Nano-Hydroxyapatite

Selenium (Se), a trace element, constitutes an essential micronutrient, possessing roles in numerous biological mechanisms. Se plays a notable part in the regulation of thyroid hormone levels [127], redox homeostasis [128], responses to inflammatory and immunological reactions [129,130,131]. Additionally, carbohydrate metabolism [132], a target metabolism for novel detection technologies in medicine [133], is reported to be highly dependent on Se balance. The trace element is reported to be responsible for upkeeping the health of the cardiovascular [134] and reproductive systems [135]. Additionally, its role in the preservation of physiological brain function [136,137] has been highly reported. Moreover, selenium acts as a cofactor, a key addition to enzymes, as it enables antioxidant enzymatic features, therefore offering protection against oxidative deterioration of the body and against cellular tissue degeneration [138,139]. Conversely, low levels of Se in the human body lead to Se deficiency, its negative effects resulting in various possible diseases [140]. Selenium supplements provide multiple benefits, such as improvement of regenerative capacity, delay of the aging process, inhibition of free radicals and prevention or treatment of endemic diseases [141,142]. It has been discovered that supplementation with selenite, an inorganic salt form of selenium, is capable to show selective toxicity and induce apoptosis in a wide range of cancerous cells and, thus, impede their proliferation processes. This implies that the supplementation with selenium can aid in the prevention and treatment of osseous tumors [137,139]. The synthesis of nano-hydroxyapatite doped with selenium was conducted by various researchers, who also evaluated the biological characteristics of the novel substances [143,144,145]. The studies concluded that the synthesis of Se-HAP led to a high degree of preosteoblast differentiation and exhibited no toxicity levels. Another study concerning the synthesis, through the process of traditional wet chemical precipitation, of hydroxyapatite doped with selenium oxyanions, revealed that low levels of hydroxyapatite supplemented with selenium were not characterized by toxicity. Moreover, Se-HAP would be able to be utilized in the treatment of bone tumors and metastases, as it displayed excellent features as a bone substitute [146].

#### 4.2.4. Strontium-Nano-Hydroxyapatite

Strontium (Sr) is a trace element, chemically similar to calcium (Ca), located in a proportion of 98% in the osseous tissue [147,148]. Research, in the form of clinical trials and experimental studies, has indicated the dual effects of stable strontium ions (${\mathrm{Sr}}^{2+}$) to promote bone formation and inhibit the bone resorptive process [149,150]. Sr ions can replace the calcium ions in the structure of hydroxyapatite, to increase the osteoblast function and to impede the proliferation of osteoclasts [151,152]. Strontium, as a substitute in the HAP structure (Figure 3), has the potential to be therapeutically used in cases of osteoporosis by stimulating the creation of novel bone tissue [153,154]. 

#### 4.2.5. Boron-Nano-Hydroxyapatite

Boron (B) is an essential microelement with multiple roles, such as molecular control of bone metabolism, steroid hormone synthesis, increase of osseous strength [156] and biomineral density (BMD) [157], which, in turn, are very important for the regulation of body calcium levels and the extracellular matrix mineralization [158]. Boron diet supplementation is capable of optimizing the metabolism of vitamin D [156], to raise BMD in rat alveolar bone [159] and to increase the concentration of serum osteocalcin (OCN) in postmenopause [160]. Studies have reported that the combination of Boron (B), nano-HAP and chitosan scaffolds facilitates the synthesis of novel bone tissue which can be successfully utilized for rat calvarial defects [158,161]. Studies have recently established the beneficial role of boron in the treatment of osteoporosis, severe osteoarthritis and rheumatoid arthritis [158]. Other studies have determined that boron facilitates the osteogenic differentiation of different cell lines, such as human hematopoietic mesenchymal stem cells, human dental stem cells, mouse preosteoblastic cellular lines and MC3T3-E1 preosteoblastic cells [162,163]. One in vivo study aimed to analyze the effect on bone healing of osteoinductive “bone-like hydroxyapatite” obtained from simulated body fluid combined with osteoinductive boron [158].

#### 4.2.6. Cobalt-Nano-Hydroxyapatite

Cobalt (Co) is a micronutrient that, when incorporated into the structure of HAP, has been demonstrated to have comparable properties to simple, native HAP [70]. Studies have determined that HAP with incorporated ${\mathrm{Co}}^{2+}$ ions stimulated the process of osteogenesis in vivo and featured significant antibacterial and antiviral action [164,165]. Additionally, it has been revealed that apatites combined with a high amount of ${\mathrm{Co}}^{2+}$ ions have a strongly positive impact on the regeneration of bone tissue affected by osteoporosis. Thus, HAP nanocrystals enriched with ${\mathrm{Co}}^{2+}$ ions can benefit from the latter’s antimicrobial and antibacterial properties in the creation process of multifunctional substances and materials with osseous and dental applications [70,160].

#### 4.2.7. Copper-Nano-Hydroxyapatite

Bone healing is comprised of several biological mechanisms; namely, inflammatory processes, vascularization and osteogenesis. Chim et al. revealed the crucial role that angiogenesis has in the reparation of bone defects [158]. Unsuccessful vascular restoration or a lack of angiogenesis can result in delays in the healing process or in failure of in vivo bone implantation [158,161]. The absence of angiogenesis leads to a slower rate of osteoid deposition and matrix synthesis, in addition to a reduction in bone healing activity [161,162]. Copper (Cu) is a trace element that has a key role in angiogenesis and facilitates the migration of endothelial cells [163]. HAP scaffolds can be combined with ${\mathrm{Cu}}^{2+}$ ions to benefit from their angiogenesis ability. Barralet et al. [166] discovered that the adsorption of low amounts of ${\mathrm{Cu}}^{2+}$ into the calcium phosphate scaffold resulted in the apparition of micro-vessels along the macro-pore axis. One study concluded that ${\mathrm{Cu}}^{2+}$ had the potential to alter HAP morphogenesis under hydrothermal conditions [167]. Thus, the inclusion of ${\mathrm{Cu}}^{2+}$ ions in the structure of HAP could positively alter its chemical and physical properties and intensify its bioactivity [164]. Another study revealed that a Cu-doped hydroxyapatite scaffold stimulated the growth and proliferation of bone mesenchymal cells (BMSCs) [165]. An additional study noted that copper-doped calcium polyphosphate (CCAP) nanocomposite scaffolds, characterized by reduced concentration of ${\mathrm{Cu}}^{2+}$ ions, lead to successful human osteoblast cellular proliferation, while the osteoblast multiplication was obstructed by nanocomposites combined with high amounts of ${\mathrm{Cu}}^{2+}$ ions [168]. The concentration of copper ions above the ideal limit results in important toxic consequences on cell activity and the appearance of free radical species that, in turn, lead to neurodegenerative diseases [169,170]. Therefore, the substitution of ${\mathrm{Cu}}^{2+}$ ions in the scaffold structure of nano-hydroxyapatite is advantageous due to its application as a bone tissue engineering nanomaterial, provided it preserves an optimal, controlled ratio of copper content [83,171].

#### 4.2.8. Silicon Nano-Hydroxyapatite

Researchers have demonstrated that the addition of silicon in nanocalcium phosphate scaffolds can constitute a procedure to promote in vitro adhesion and proliferation in osteoblasts [172].

#### 4.2.9. Multi-Substituted Hydroxyapatite

Using Collagen 1 molecules (COL), researchers have functionalized multi-substituted HAP (ms-HAP) HAP-1.5 wt% Mg-0.2 wt% Zn-0.2 wt% Si nanoparticles (NPs), giving rise to a core/shell NP prototype (ms-HAP/COL). Next, a complex ms-HAP/COL@PLA/COL composite was obtained by additional embedment into a polylactic acid (PLA) matrix and finally covered with COL layers. Testing revealed the important osseointegration of implants as well as improved bone regeneration [173]. Similarly, nano-hydroxyapatite (HAP) substituted with multiple cations (Sr^2+^, Mg^2+^ and Zn^2+^) demonstrated a measurable in vitro ion release [174]. Moreover, novel-generation multi-substituted hydroxyapatites (ms-HAPs) have been recently reported, with important potential for bone regeneration. Bone healing was enhanced by the enhancement of properties exerted by both HAP and functional elements [175]. An interesting study reported multi-substituted hydroxyapatite (ms-HAP) functionalized with collagen (ms-HAP/COL), embedded into a poly-lactic acid (PLA) matrix (ms-HAP/COL@PLA) and subsequently covered with self-assembled COL layer (ms-HAP/COL@PLA/COL, named HAPc). The so-designed implants demonstrated superior bone consolidation in the presence of high-frequency pulsed electromagnetic short-wave (HF-PESW) exposure [176]. Other approaches included doping nano-hydroxyapatite with silver (0.25 wt%), zinc (0.2 wt%) and gold (0.025 wt%) by means of an innovative wet chemical approach, jointly with a reduction procedure for gold and silver [177].

#### 4.2.10. Hydroxyapatite Combined with Other Ions

Manganese (Mn), an essential trace element, was revealed to aid in the regeneration of bone tissue, as it is capable of promoting the proliferation of osteoblasts and activating the osteoblast metabolic processes [178]. Other studies reported that the combination between the crystal structure of HAP and iron (Fe) was able to add magnetic properties to the HAP scaffold and, thus, to stimulate the remodeling and regeneration of bone tissue [179,180]. Additionally, the super-paramagnetic nature of Fe-doped HAP and its capacity for drug delivery have been demonstrated to be osteoblastic activity enhancers, therefore positively impacting osseous regeneration [181,182,183]. Extensive bone growth and development, as well as a higher degree of implant coverage by the bone tissue, have also been reported for HAP combined with Silicon [184]. HAP imbued with small amounts of Silver (Ag) ions was deemed to be an advantageous antibacterial nanomaterial for use in dental and orthopedic implants [171,185]. Comparable antibacterial properties have been reported for HAP structures doped with Caesium (Ce) or Europium (Eu) ions [171,186,187]. A study reported that, in white rabbits, the phosphate (${\mathrm{PO}}_{4}^{3-}$) group, substituted with the carbonate (${\mathrm{CO}}_{3}^{2-}$) group, displayed superior biological integration of HAP implants [188]. Adding Hafnium (Hf) ions to HAP provided it with the ability to create, under ionizing radiation, high amounts of reactive oxygen species, with possible applications in photodynamic anti-tumor treatments [189]. Table 3 provides a list of ions that can be added in the structure of hydroxyapatite, as well as their roles and advantages of supplementation.

## 5. Hydroxyapatite Enriched with Curcumin

Curcumin, a low-molecular-weight polyphenol derived from Curcuma longa that represents the active component of turmeric, has been the subject of extensive research due to its beneficial physiological characteristics [191]. Curcumin possesses a wide array of pharmacological properties; namely, anti-inflammatory, anti-infection, anti-oxidation, anti-coagulation, anti-carcinogenic, anti-liver fibrosis and anti-atherosclerosis, etc. Moreover, curcumin can positively influence the treatment of osteoporosis and can aid in the fracture healing process, implant repair and orthodontic treatment as well [191]. Studies conducted on mice revealed that curcumin supplementation led to improved bone microarchitecture and increased mineral density [192]. The effect of curcumin in the preservation of bone integrity and augmentation of mineral bone density in the lumbar vertebrae in ovariectomized (OVX) rats has been reported in the literature [193,194,195]. Another study noted that curcumin has a role in bone loss prevention in OVX mice. Additionally, curcumin can impede the synthesis of mature osteoclasts, thereby enhancing its bone-protective property [196,197]. The beneficial role of curcumin on bone tissue development and regeneration is owed to its ability to impede H_2_O_2_-stimulated osteoblast apoptosis [198], thus improving the osteoblast mitochondrial activity [199] and recovering the osteogenic differentiation of osteoblast and bone-marrow-derived mesenchymal stem cells (BMSCs), impaired by high concentrations of glucose [200,201,202,203]. Several in vitro, in vivo, and preclinical studies have concluded that curcumin plays a significant part in the regulation of various cell-signaling pathways and exhibits a notable cytotoxic potential in relation to different tumor cells [191]. Studies have also reported that curcumin possesses osteogenic and chemopreventive potential, which are significant factors for the prevention of osteosarcoma [191,204]. Other in vitro and in vivo reports explain the essential mechanisms and pathways of curcumin-induced cellular apoptosis in bone cancer, namely the inhibition of nuclear factor-κβ (NF-κβ) and the expression of interleukin-6 (IL-6) and interleukin-11 (IL-11), the interference with tumor-induced angiogenesis by the downregulation of vascular endothelial growth factor and matrix metalloproteinases-9 (MMP-9) and the prevention of the expression of extracellular signal-regulated kinase [204,205,206]. An in vivo study revealed that, in rats, the combined therapy between curcumin and cisplatin, a chemotherapy medication, showed superior control over the tumor marker of the cancerous cells in comparison to normal cells [207]. A study in mice found that a dual treatment strategy, consisting of curcumin administration and radiotherapy, lead to an increased rate of apoptosis of cancer cells and diminished radioresistance [208]. The MTC group has also recently published an extensive review on curcumin-enriched HAP [209]. Another study presented the creation of a bifunctional bone tissue engineering scaffold, that was capable of impeding the proliferation of malignant bone cells and of promoting the synthesis of healthy bone cells in the porous scaffold structure, as a possible therapeutic approach for bone defects following tumor resection [205].

Hydroxyapatite can be utilized as a carrier for the prolonged release of curcumin nanoparticles (Cur-NPs) (Figure 4). HAP nanoparticles present increased affinity towards hydrophobic and hydrophilic drugs, proteins and nucleic acids, and are able to cross cellular membranes [210]. For successful medication delivery into target tumor cells, in optimal concentrations, HAP nanoparticles feature a high capacity for drug loading and adequate incorporation inside cells. A noteworthy possibility is the use of hollow nanoporous HAP nanoparticles, with large void fractions contained inside the hollow interior, that can function as drug storage locations. Conversely, the nanoporous shell has the role of a permeable barrier, in order to impede the burst release of encapsulated drugs [211,212,213,214]. 

## 6. Conclusions and Future Perspectives

In this review, the most recent trends in bone tissue regeneration have been presented, with emphasis on hydroxyapatite, its characteristics, interactions with bone components and its combinations with different substances. In summary, hydroxyapatite is an excellent material for bone synthesis and repair due to its biocompatibility, bioactivity, osteoinductivity and osteoconductivity. Its role in bone cell activity, as well as its importance in bone-signaling pathways through cellular and molecular mechanisms, has been outlined. Nanoparticles that consist of hydroxyapatite combined with various ions have shown many advantages for bone growth, proliferation and regeneration, each acting according to different mechanisms. Additionally, studies on hydroxyapatite enriched with natural products, namely, curcumin, have shown promising results, having numerous applications in the treatment of bone cancer, bone defects and in the incorporation of drugs inside bone cells. In the future, nanoscale biological response mechanisms of cells to different surfaces will create interesting research topics. It is the authors’ belief that the biological processes of bone growth, development and remodeling present a wide array of opportunities for further research, which can also lead to positive developments in the design of hydroxyapatite biomaterials for accelerating osseous regeneration.

## Figures and Tables

**Figure 1 materials-16-01303-f001:**
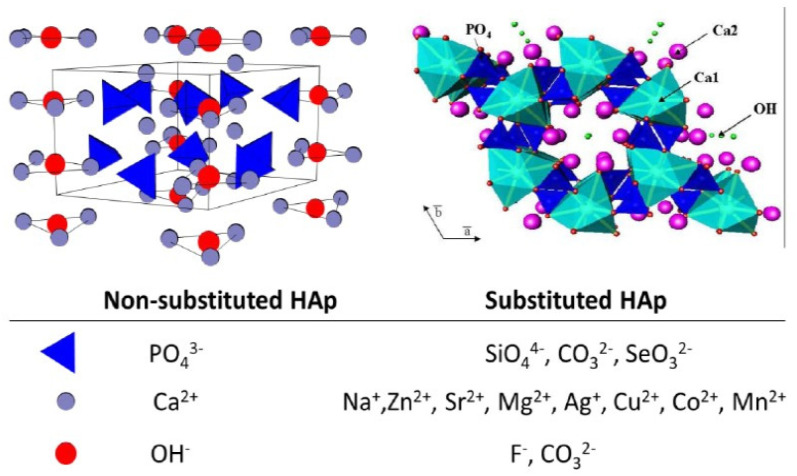
Simple, non–substituted hydroxyapatite and its structure (**left**) and possible ionic substitutions in the structure of hydroxyapatite (**right**). Reproduced with permission from Arcos Daniel et al., 2021 [91].

**Figure 2 materials-16-01303-f002:**
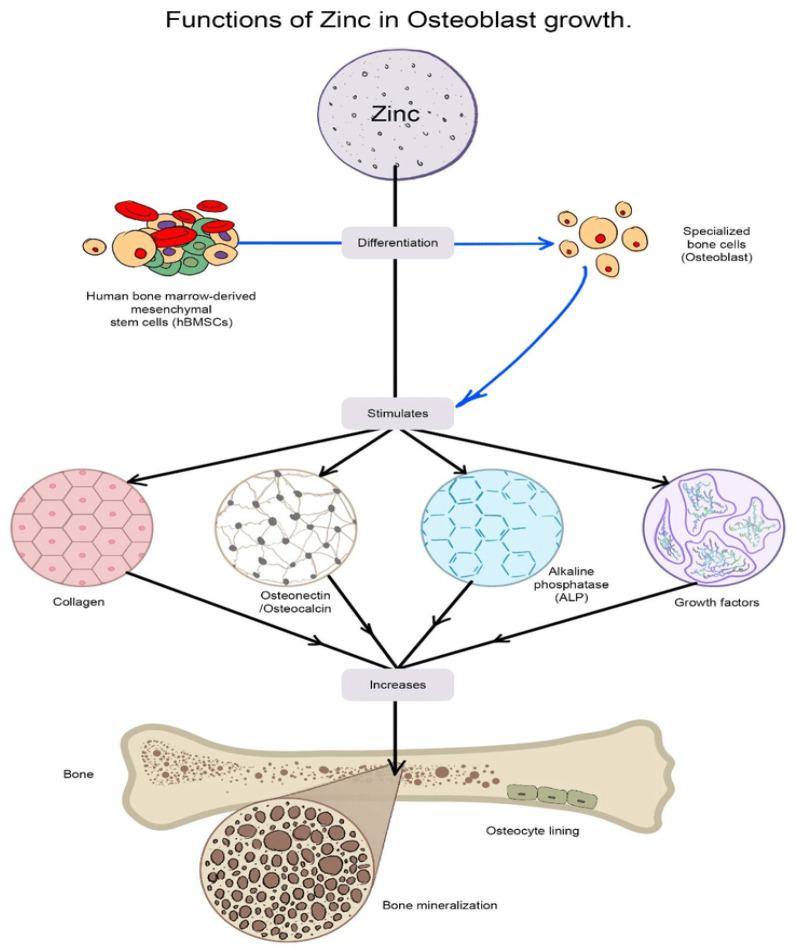
The roles of zinc in osteoblast growth. Zinc stimulates the gene expression of type 1 collagen, osteocalcin and alkaline phosphatase, and increases the production of growth factors in osteoblasts. Reproduced with permission from Mandal, A.K. et al., 2022 [126].

**Figure 3 materials-16-01303-f003:**
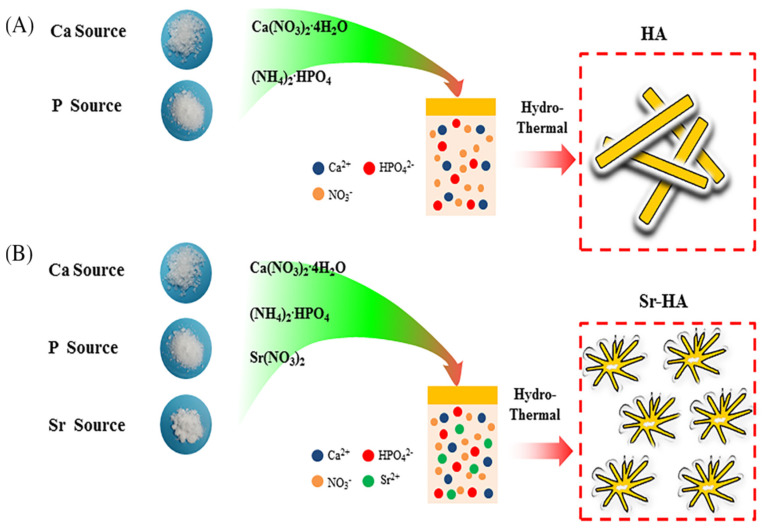
Hydroxyapatite (HAP) synthesis from the Ca and P sources (**A**), and strontium–hydroxyapatite (Sr–HAP) synthesis from the Ca, P and Sr sources (**B**), using the wet chemical method. Reproduced with permission from Jingfeng Li et al., 2018 [155].

**Figure 4 materials-16-01303-f004:**
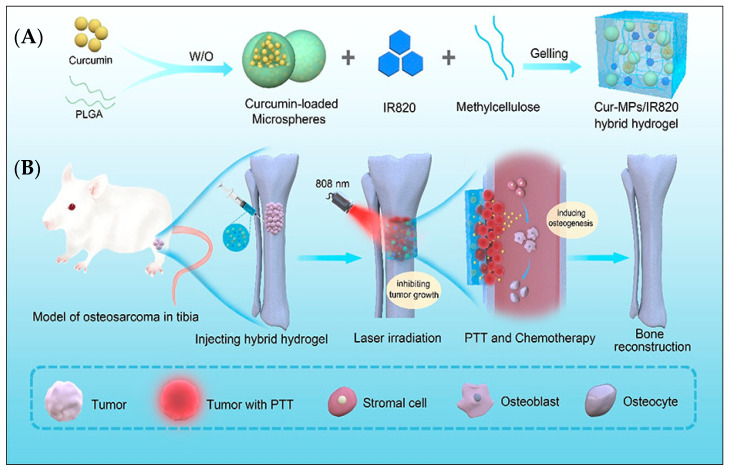
(**A**) Preparation scheme for the Cur-MP/IR820 hybrid hydrogel. (**B**) Illustration of Cur-MP/IR820 hybrid hydrogel applied in osteosarcoma chemo-photothermal combined therapy and following bone reconstruction. Reproduced with permission from Bowen Tan et al., 2021 [215].

**Table 3 materials-16-01303-t003:** Hydroxyapatite combined with ions: roles of ions and supplementation advantages.

Ions Added to nHAP	Ion Roles	Advantages of HAP Ion Supplementation	Reference
Magnesium	Cartilage and bone structure Skeletal metabolism Osteoblast and osteoclast activity	Superior mineralization Improved osseointegration Superior morphology, granulation, composition, solubility and crystallinity Biocompatibility	[105,114] [70] [113]
Zinc	Bone metabolism stimulation Positive impact on osteoblast and osteoclast activity Decrease of early bone resorption and deterioration	Promotes osteogenesis in osteoporotic bones Osteoregeneration Anti-inflammatory ability	[114] [120,121,122] [115,123]
Selenium	Regulation of thyroid hormone levels Redox homeostasis Inflammatory and immunological reaction response Carbohydrate metabolism Cardiovascular health Reproductive health Physiological brain function Enzyme cofactor and protection against tissue deterioration	Regeneration Aging delay Free radical inhibition Endemic diseases prevention and treatment Selective toxicity and apoptosis of cancer cells Bone tumor and metastasis prevention and treatment Preosteoblast differentiation No toxicity	[127,128,129,130,131,132,133,134,135,136,137,138,139]
Strontium	Bone synthesis Inhibition of bone resorption	Increased osteoblast function Impeded osteoclast proliferation Osteoporosis treatment	[149,150,151,152,153,154]
Boron	Bone metabolism molecular control Synthesis of steroid hormone Bone strength and biomineral density (BMD) increase	Vitamin D metabolism optimization Treatment of calvarial defects in rats Osteoporosis, osteoarthritis and rheumatoid arthritis treatment Osteogenic differentiation of cell lines	[156,157,158,190] [158,161,162,163]
Cobalt	Bone tissue regeneration Antibacterial and antiviral action	Stimulation of in vivo osteogenesis	[160] [164,165]
Copper	Angiogenesis Endothelial cell migration	Synthesis of micro-vessels Positive impact on chemical and physical properties of HAP Improvement of HAP bioactivity Growth and proliferation of bone mesenchymal cells Osteoblast proliferation (reduced Copper ion concentrations)	[163,164,165,166,167,168] [83,171]
Silicon	Bone growth and development	Promotion of in vitro osteoblast adhesion and proliferation	[172,184]
Manganese	Bone tissue regeneration	Osteoblast proliferation promotion and metabolism activation	[178]
Iron	Magnetic properties Drug delivery	Stimulation of bone tissue remodeling and regeneration Enhancement of osteoblastic activity	[179,180]
Silver Caesium Europium	Antibacterial properties	Use in dental and orthopedic implants	[171] [185,186,187]
Carbonate		Superior bio-integration of HAP implants	[188]
Hafnium	Creation of reactive oxygen species	Photodynamic anti-tumor treatment	[189]

## Data Availability

Not applicable.

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
