# Peer review of "Recent Trends in Hydroxyapatite Supplementation for Osteoregenerative Purposes"

_materials, 2023, doi:10.3390/ma16031303_

Round 1

Reviewer 1 Report

Please, modify the abstract, because it seems more like a conclusion.

For each section, it would be useful to show a comparative table or graph as a summary of the studies indicated.

The conclusions are too brief. A summary of each section discussed should be indicated as well as a personal opinion of the authors on hydroxyapatite supplementation for osteoregenerative purposes

Reviewer 2 Report

General comments

The article Recent trends in hydroxyapatite supplementation for osteoregenerative purposes is well written and well organized. In this review, the outline the cellular and molecular roles of hydroxypatite, its interactions with bone cells, as well as its nano-combinations with various ions and natural products and their effects on bone growth, development and repair. There is no description of the methodology, the keywords adopted, the criteria for selection and rejection of papers. The authors present a review paper, and the literature list contains 216 items. Conclusions are presented in a concise manner.  In my opinion, it seems necessary to make minor corrections to allow a more thorough understanding of the topic. The following are my comments.

Minor comments:

Abstract:

The abstract is too short, please expand it with the most important information regarding the literature review presented.

Introduction

Please add to the introduction brief information on typical materials used to fill bone defects such as bone cements. Below are some works that may be useful to the authors.

- https://doi.org/10.3390/ma15062197

- https://doi.org/10.3390/ma15165577

- https://doi.org/10.1023/B:JMSM.0000046387.70323.41

Materials and Methods

There is no description of the methodology, the keywords adopted, the criteria for selection and rejection of papers. I recommend adding a short section containing the mentioned information, this will allow easier reception of the article.

After making the appropriate additions, the article may be accepted for publication.
